# Prevalence, Incidence, and Outcomes of Hyperkalaemia in Patients with Chronic Heart Failure and Reduced Ejection Fraction from a Spanish Multicentre Study: SPANIK-HF Design and Baseline Characteristics

**DOI:** 10.3390/jcm11051170

**Published:** 2022-02-22

**Authors:** Juan F. Delgado-Jiménez, Javier Segovia-Cubero, Luis Almenar-Bonet, Javier de Juan-Bagudá, Antonio Lara-Padrón, José Manuel García-Pinilla, Juan Luis Bonilla-Palomas, Silvia López-Fernández, Sonia Mirabet-Pérez, Inés Gómez-Otero, Antonio Castro-Fernández, Beatriz Díaz-Molina, Josebe Goirigolzarri-Artaza, Luis Miguel Rincón-Díaz, Domingo Andrés Pascual-Figal, Manuel Anguita-Sánchez, Javier Muñiz, María G. Crespo-Leiro

**Affiliations:** 1Instituto de Investigación y Servicio de Cardiología del Hospital Universitario 12 de Octubre, 28041 Madrid, Spain; javierdejuan166@hotmail.com; 2Centro de Investigación Biomédica En Red Cardiovascular (CIBERCV), 28029 Madrid, Spain; jsecu@telefonica.net (J.S.-C.); lualmenar@gmail.com (L.A.-B.); marlucale41@gmail.com (J.M.G.-P.); smirabet@santpau.cat (S.M.-P.); maria.ines.gomez.otero@sergas.es (I.G.-O.); beadimo@gmail.com (B.D.-M.); josebegoiri@gmail.com (J.G.-A.); lmrincondiaz@hotmail.com (L.M.R.-D.); dpascual@um.es (D.A.P.-F.); marisacrespo@gmail.com (M.G.C.-L.); 3Servicio de Cardiología, Hospital Universitario Puerta de Hierro, 28222 Madrid, Spain; 4Servicio de Cardiología, Hospital Universitario La Fe, 46026 Valencia, Spain; 5Servicio de Cardiología, Hospital Universitario de Canarias, 38320 Santa Cruz de Tenerife, Spain; alarapadron@gmail.com; 6Servicio de Cardiología, International Business Information Management Association, Hospital Universitario Virgen de la Victoria, 29010 Malaga, Spain; 7Servicio de Cardiología, Hospital San Juan de la Cruz, Úbeda, 23400 Jaen, Spain; jnlsbnll@hotmail.com; 8Servicio de Cardiología, Hospital Universitario Virgen de las Nieves, 18014 Granada, Spain; silvialopezf@msn.com; 9Servicio de Cardiología, Hospital de la Santa Creu i Sant Pau, 08041 Barcelona, Spain; 10Servicio de Cardiología, Complexo Hospitalario Universitario de Santiago de Compostela, 15706 A Coruña, Spain; 11Servicio de Cardiología, Hospital Universitario Virgen de la Macarena, 41009 Seville, Spain; antoniocastro007@gmail.com; 12Servicio de Cardiología, Hospital Universitario Central de Asturias, 33011 Oviedo, Spain; 13Servicio de Cardiología, Hospital Universitario Clínico de San Carlos, 28040 Madrid, Spain; 14Servicio de Cardiología, Hospital Ramón y Cajal, 28034 Madrid, Spain; 15Servicio de Cardiología, Hospital Clínico Universitario Virgen de la Arrixaca, 30120 Murcia, Spain; 16Servicio de Cardiología, Hospital Universitario Reina Sofía, 14004 Cordoba, Spain; manuelanguita@secardiologia.es; 17Instituto de Investigación Biomédica de A Coruña, Universidade da Coruña (UDC), 15006 A Coruña, Spain; proyectos@odds.es; 18Servicio de Cardiología, Complexo Hospitalario Universitario, 15006 A Coruña, Spain

**Keywords:** hyperkalaemia, heart failure, medical treatment

## Abstract

Hyperkalaemia is a growing concern in the treatment of patients with heart failure and reduced ejection fraction (HFrEF) as it limits the use of some prognostic-modifying drugs and has a negative impact on prognosis. The objective of the present study was to estimate the prevalence of hyperkalaemia in outpatients with HFrEF and its impact on achieving optimal medical treatment. For this purpose, a multicentre, prospective, and observational study was carried out on consecutive HFrEF patients who were monitored as outpatients in heart failure (HF) units and who, in the opinion of their doctor, received optimal medical treatment. A total of 565 HFrEF patients were included from 16 specialised HF units. The mean age was 66 ± 12 years, 78% were male, 45% had an ischemic cause, 39% had atrial fibrillation, 43% were diabetic, 42% had a glomerular filtration rate < 60 mL/min/1.7 m^2^, and the mean left ventricular ejection fraction was 31 ± 7%. Treatment at the study entry included: 76% on diuretics, 13% on ivabradine, 7% on digoxin, 18.9% on angiotensin-conversing enzyme inhibitors (ACEi), 11.3% on angiotensin receptors blockers (ARBs), 63.8% on angiotensin-neprilysin inhibitors (ARNi), 78.5% on mineralocorticoid receptor antagonists (MRAs), and 92.9% on beta-blockers. Potassium levels in the baseline analysis were: ≤5 mEq/L = 80.5%, 5.1–5.4 mEq/L = 13.8%, 5.5–5.9 mEq/L = 4.6%, and ≥6 mEq/L = 1.06%. Hyperkalaemia was the reason for not prescribing or reaching the target dose of an MRAs in 34.8% and 12.5% of patients, respectively. The impact of hyperkalaemia on not prescribing or dropping below the target dose in relation to ACEi, ARBs, and ARNi was significantly less. In conclusion, hyperkalaemia is a frequent problem in the management of patients with HFrEF and a limiting factor in the optimisation of medical treatment.

## 1. Introduction

In recent years, multiple reasons have contributed to an increased disease burden from heart failure (HF) in industrialised countries [1]. It is foreseeable that the incidences of this important health problem will continue increasing [1,2]. In parallel with this development, several treatments already included in clinical practice guidelines have proved to be effective in select groups of patients, especially those with HF and reduced left ventricular ejection fraction (HFrEF) [3]. Renin-angiotensin-aldosterone system inhibitors (RAASi) and beta-blockers (BB), which have a class IA indication for patients with HFrEF in clinical practice guidelines, increase hyperkalaemia risk [4]. The occurrence of hyperkalaemia often limits RAASi use and/or leads to dose reduction and discontinuation, thereby reducing their potential benefits [4,5].

In HFrEF, the relationship between potassium (K^+^) concentration and adverse outcomes appears to be U-shaped, where both low- and high-K^+^ levels are associated with negative effects [6,7]. However, it remains unclear to what extent dyskalaemia is a risk factor itself rather than a risk marker representing the patients’ overall clinical status, other comorbidities, and/or use or non-use of HF medication [8].

Hyperkalaemia is associated with conduction disorders and carries the risk of potentially life-threatening arrhythmias [4,9,10]. The threshold risk for the development of hyperkalaemia-associated arrhythmic emergencies and death varies widely between patients; the safe serum K^+^ zone is not well established [4].

Nonetheless, there is some limited prospective evidence for the estimation of hyperkalaemia in HFrEF patients. Especially in a recent cohort of well-treated patients in specialised units and once the treatment has been optimised.

So, the main objective of the SPANIK-HF study (Spanish multicenter study on prevalence, incidence, and prognosis of hyperkalaemia in patients with HF) was to estimate the prevalence and incidence of hyperkalaemia during the first year of follow-up, its relationship with suboptimal treatment and clinical outcomes (mortality and hospital admission) in a recent cohort of HFrEF outpatients under optimised medical therapy.

This first article will address data from the reference population, prevalence of hyperkalaemia, and interference with optimisation of medical treatment at enrolment.

## 2. Materials and Methods

### 2.1. Study Design

The SPANIK-HF trial (ClinicalTrials.gov NCT_NCT04141800) is a national multicentric prospective observational study that includes consecutive HFrEF patients in the outpatient setting. An inclusion baseline visit and a follow-up visit at 12 months were scheduled for collecting clinical and blood sample data. The study was conducted under ordinary conditions of clinical practice, i.e., no additional procedures or interventions were performed.

Prevalence of hyperkalaemia (as defined below) at baseline was determined as well as the appearance of new hyperkalaemia cases among patients that had normal K^+^ levels at the baseline visit but were hyperkalaemic at follow-up; the relationship of the latter group with outcomes of interest was also determined.

### 2.2. Definition of the Study Population and Selection Criteria

The study population was composed of HF outpatients with HFrEF from 16 Spanish centres between April–July 2019. These centres were selected from either (i) excellence-certified HF units by the Spanish Society of Cardiology in its program SEC HF-excellence [11]; (ii) included in the Centro de Investigación en Red en Enfermedades Cardiovasculares (CIBERCV) consortium; or (iii) with good performance in previous similar registries. The selection among these groups was not random but based on interest and high-performance criteria in the field of HF.

### 2.3. Inclusion Criteria

Patients, women or men, aged 18 or older with documented HFrEF (LVEF ≤40%) diagnosis and who signed written informed consent.

### 2.4. Exclusion Criteria

Exclusion criteria were: any type of disorder affecting the capacity to give informed written consent; clinical trial enrolment at the moment of the inclusion; patients suffering stage 5 chronic kidney disease; and patients predicted to have less than one year of life span due to diseases different from HF and not having completed HF drug titration stage at the time of inclusion (this stage is not complete if, on a doctor’s judgement, possible maximum doses had not been reached for RAASi medications and any of these drugs had been included or dose-modified in the recruitment visit).

At any time, patients could leave the study (and would be censored in the analysis) or retire their consent (and would be excluded from the analysis).

### 2.5. Main Outcome Variables

The main outcome variables for the present study were: (i) hyperkalaemia—serum K^+^ was measured at the baseline and follow-up visits as well as at any intermediate hospital admissions; and (ii) proportion of patients using drugs with class I indication in clinical practice guidelines (ACEi, ARBs, ARNi, and MRAs) and proportion of patients using optimal doses of these drugs. The reasons for not using these classes of drugs, or not receiving the optimal doses, were recorded.

### 2.6. Definition of Hyperkalaemia

In the absence of universally accepted limits regarding the severity of hyperkalaemia, in this study hyperkalaemia was defined as serum K^+^ > 5 mmol/L and was classified as mild (>5.0 to <5.5 mmol/L), moderate (5.5 to <6.0 mmol/L), or severe (≥6.0 mmol/L).

### 2.7. Considerations about Sample Size

The sample size was calculated for the objective of the association of hyperkalaemia with clinical events during follow-up. Assuming a 10% risk of outcomes (mortality or HF hospital admission) among the exposed (hyperkalaemia), 2.5% among the non-exposed, and a 7:1 non-exposed/exposed ratio, 73 exposed and 511 non-exposed patients should be included to have an 80% power to detect these differences with a 95% confidence level. To allow for incidences where patients fall out of the study, a recruitment goal was set at 600 patients. This sample size allowed a precision of ±2.4%, assuming a prevalence of hyperkalaemia at baseline of 10% and with a 95% confidence level. Furthermore, assuming 5% hyperkalaemia cumulative incidence through follow-up among hyperkalaemia-free patients at baseline, the proposed sample size allowed a precision of ±1.8% with a 95% confidence level.

### 2.8. Data Collection

Every researcher explained the information sheet to prospective patients that might fulfil the inclusion criteria (and did not present any exclusion criteria). Researchers then asked them if they were willing to sign the informed consent. Researchers were required to warrant the accuracy and completion of the data collected for the study. Data registered at the electronic Case Report Form (eCRF) should be consistent with source documents used for their collection.

### 2.9. Data Management

Data were collected during initial and follow-up visits and integrated into a single database on the web platform. Researchers are responsible for the information included in the database, which is secured by personal login and password. The online platform includes ranges and rules to minimise data logging errors.

## 3. Results

### 3.1. Baseline Patient Characteristics

The patient population included is HFrEF outpatients, asymptomatic or with moderate chronic symptoms, under optimal medical treatment for HFrEF. The average patient in the study population was a 65.9 year-old male, stable under medical treatment, with a heart rate below 70 bpm, and a systolic blood pressure in the range of 100–140 mmHg. Usual comorbidities of atrial fibrillation and diabetes mellitus were observed, as well as 41.5% with glomerular filtration rate below 60 mL/min/1.7 m^2^ (Table 1).

### 3.2. Medication at the Time of Inclusion at the Baseline Visit

Only 18.1% of participants received sodium/glucose cotransporter-2 inhibitors (SGLT2i). They did so as an antidiabetic because at the time of recruitment the benefit of this pharmacological group in the HFrEF of the non-diabetic population was not known (Table 2).

### 3.3. Distribution of K^+^ Values at the Baseline Visit

The severity of hyperkalaemia was classified as either mild (>5.0 to <5.5 mmol/L), moderate (5.5 to <6.0 mmol/L), or severe (≥6.0 mmol/L). Some degree of hyperkalaemia was present in almost 20% of the patients at the time of inclusion, being mild in 13.8%, moderate in 4.6%, and severe in 1.1% of participants (Table 3).

### 3.4. Interference of Hyperkalaemia in the Optimisation of Medical Treatment

The proportion of patients using drugs with class I indication in clinical practice guidelines (i.e., ACEi/ARBs/ARNi, MRAs, and BBs) can be considered adequate, considering that 92.9% have BBs, 94% have ACEi or ARBs or ARNi, and 78.5% have MRAs (Table 4).

In the analysis of how hyperkalaemia could interfere with the use of prognostic-modifying drugs, it was observed that of the 6% of patients who did not receive ACEi, ARB, or ARNi, about 9% did not receive a drug from this group due to contraindication or intolerance related to hyperkalaemia. Hyperkalaemia was the reason in a third of patients who did not receive MRAs. However, the percentage of patients (60–70%) who do not reach the recommended target doses according to clinical practice guidelines is a relatively more important parameter to consider.

When participants were asked why the target dose had not been reached for each drug administered, hyperkalaemia was the reason for <5% of those on ACEi and ARBs, 5.8% for those on ARNi, and 12.5% for those on MRAs.

## 4. Discussion

Hyperkalaemia is a problem that has not been given sufficient attention and may have a more significant role than has previously been thought in those HF patients that have not been receiving the treatments indicated for their condition or at the demonstrated efficient doses. For this observational, prospective, multicenter study, we examined hyperkalaemia in a specific cohort of consecutive outpatients with HFrEF from HF units who, in the opinion of their HF specialist, were receiving appropriate medical treatment.

The main findings of this first part of the baseline data of our study show that hyperkalaemia, even to a mild degree, was present in almost 20% of the patients. Second, in a subgroup of patients, we found that hyperkalaemia was the alleged reason for not reaching the target dose of treatments with a class I indication, especially in the case of MRAs.

The prevalence of hyperkalaemia in another large Spanish population of patients with chronic HFrEF, included in the European Society of Cardiology Heart Failure Long-Term Registry (ESC-HF-LT-R), was 16.06% [12], and therefore slightly lower than the 19.5% reported here. Although the prevalence is similar between both studies, the higher prevalence in the current study compared to the ESC-HF-LT-R study cohort could be justified because the current study required that the patient be optimised while in the ESC-HF-LT-R study, the patient could be in the drug titration phase.

The recommended medical treatment of HFrEF patients has therapeutic targets which allow better prognosis [3]. The quality of this medical treatment is excellent in the context of a clinical trial [13]. However, analysis of clinical registries and databases indicates that there is room for improvement [14,15,16].

In a recent study, the analysis of medical treatment for HFrEF in a Spanish population showed that 35.7% of patients were not prescribed BBs, 19.5% were not prescribed ACEi/ARBs or ARNi, and 70.2% were not prescribed MRAs [1]. However, the European Society of Cardiology treatment guidelines for HFrEF, in effect at that time, recommended an ACEI or ARB or ARNi, a BB, and an MRA (indication class I, level of evidence A or B for SAC/VAL) [17].

The high non-prescription rates found here thus demonstrate that treatment optimisation should be prioritised as an essential step toward reducing symptoms, hospitalisation, and mortality and increasing functional capacity and quality of life. The reasons for the non-prescription and underdosing of evidence-based therapies are usually not apparent from a detailed review of the medical history. Many times, this deficiency can result from the therapeutic inertia that is based on the “stability” of the patient, whereas for other cases it may result from intolerance or contraindications for different reasons.

Hyperkalaemia in HF is often associated with the use of ACEi, ARBs, MRAs, including the use of ARNi, and also in those with older age, diabetes mellitus, and chronic kidney disease [18]. The occurrence of hyperkalaemia often limits RAASi use and/or can lead to dose reduction and discontinuation, thereby reducing their potential benefits. Thus, hyperkalaemia is a major concern for clinicians, particularly in association with the use of MRAs [19]. In a recent observational study including patients initiating MRA therapy, the occurrence of hyperkalaemia led to MRA discontinuation in 47% and dose reduction in 10% of patients. Once MRA was discontinued, more than three-fourths of patients were not re-started on MRAs during the subsequent year [20].

The findings of our study pointed in a similar direction. In this particular scenario of patients under treatment considered optimised, with a high proportion of participants on ARNi (63.8%), hyperkalaemia, due to either intolerance or contraindication, is still a reason for not prescribing or reaching the target dose of an ARM in 34.8% and 12.5% of patients, respectively. The impact of hyperkalaemia on not prescribing or prescribing below the target dose, in relation to ACEi, ARBs, and ARNi, was significantly less. The recent availability of safe and tolerable gastrointestinal K^+^ binders (patiromer and sodium zirconium cyclosilicato) allows for chronic management of hyperkalaemia and may enable RAASi therapy optimisation and will most likely change the current scenario.

The results presented here refer to the baseline visit exclusively. As the temporal relationship between discontinuation of drugs and potential causes of hyperkalaemia is sometimes difficult to elucidate, the follow-up information from this study will help to disentangle the relationship among hyperkalaemia, other comorbidities (such as renal disfunction), and discontinuation of drugs of proven efficacy in HFrEF patients.

## 5. Conclusions

In conclusion, hyperkalaemia is a frequent problem in the management of patients with HFrEF. Likewise, it is a limiting factor in the optimisation of medical treatment, especially for MRAs but also to some degree for ACEi, ARBs, and ARNi.

## Figures and Tables

**Table 1 jcm-11-01170-t001:** Baseline patient characteristics at the time of inclusion.

Patient Characteristic	Value
Age ([mean (SD)] in years)	65.9 (12.3)
<60 years	27.4
60–69 years	30.8
70–79 years	28.5
≥80 years	13.3
Male	78.2
Ischemic aetiology	45.3
Body mass index (*n* = 565; [mean (SD)] in kg/m^2^)	28.6 (4.8)
Heart rate ([mean (SD)] in bpm)	68.2 (12)
Systolic blood pressure ([mean (SD)] in mmHg)	115.0 (17.9)
Diastolic blood pressure ([mean (SD)] in mmHg)	69.1 (10.4)
Atrial fibrillation	38.8
Diabetes mellitus	43.4
Stroke or transient ischemic attack	9.7
Chronic obstructive pulmonary disease	12.9
Chronic kidney disease	26.4
Cancer	5.8
Depression	10.4
Left ventricular ejection fraction [mean (SD)]	30.7 (7.3)
Glomerular filtration rate (MDRD) (*n* = 564; [mean (SD)] in ml/min/1.73 m^2^)	67.6 (25.8)
<30 mL/min/1.73 m^2^	5.3
30–59 mL/min/1.73 m^2^	36.2
≥60 mL/min/1.73 m^2^	58.5
Atrial peptides	
BNP (*n* = 37; [mean (SD)] in pg/mL)	466.76 ± 481.56
NT-proBNP (*n* = 490; [mean (SD)] in pg/mL)	2474.1 (3963.6)

Notes: All values expressed in % unless otherwise indicated (*n* = 565). MDRD, Modification of Diet in Renal Disease equation.

**Table 2 jcm-11-01170-t002:** Medication at the time of inclusion at the baseline visit.

Medication	%
Diuretics	76.3
Ivabradine	12.9
Digital	7.1
Statins	70.8
SGLT2i	18.1
Antiplatelet	40.0
Anticoagulants	48.5
Amiodarone	11.7
Nitrates	6.2

Notes: Specific drugs for the treatment of heart failure and reduced ejection fraction (HFrEF) are not included. Results are expressed in % (*n* = 565). SGLT2i, Sodium/glucose cotransporter-2 inhibitors.

**Table 3 jcm-11-01170-t003:** Distribution of K^+^ values and prevalence of hyper- and hypokalaemia at the baseline visit.

	Total (*n*)	Mean (SD)	IC95 (%)	
K^+^	565	4.6 (0.52)		
Distribution of K^+^		%		
≤5 mEq/L	455	80.5	77.0	83.7
5.1–5.4 mEq/L	78	13.8	11.1	16.9
5.5–5.9 mEq/L	26	4.6	3.0	6.7
≥6 mEq/L	6	1.1	0.39	2.3
Hypokalaemia (<3.5 mEq/L)	12	2.1	1.1	3.7

**Table 4 jcm-11-01170-t004:** Specific drugs for the treatment of HFrEF and the importance of hyperkalaemia as the reason for not administering or reaching a target dose.

Drugs	Number of Patients Receiving Treatment [*n* (%)]	Hyperkalaemia as a Cause of Contraindication or Intolerance [*n*/Number of Patients with Contraindication or Who Did Not Tolerate the Drug (%)]	Patients Not Reaching Target Dose [*n*/Number Patients Receiving Drug (%)]	Hyperkalaemia as a Cause of Not Reaching Target Dose [*n*/Number of Patients Who Do Not Reach the Target Dose (%)]
ACEi	107 (18.9)	3/72 (4.2)	67/107 (62.6)	2/67 (3.0)
ARBs	64 (11.3)	2/83 (2.4)	46/64 (71.9)	2/46 (4.3)
MRAs	441 (78.5)	24/69 (34.8)	311/441 (70.5)	39/311 (12.5)
ARNi	361 (63.8)	3/65 (4.6)	225/361 (62.3)	13/225 (5.8)
BBs	525 (92.9)	N.A.	335/525 (63.8)	N.A.

Notes: ACEi, angiotensin-converting enzyme inhibitors; ARBs, angiotensin receptor blockers; ARNi, angiotensin receptor-neprilysin inhibitors; BBs, beta-blockers; HFrEF, heart failure and reduced ejection fraction; MRAs, mineralocorticoid receptor antagonists; N.A., not applicable.

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
