# Peer review of "Prevalence, Incidence, and Outcomes of Hyperkalaemia in Patients with Chronic Heart Failure and Reduced Ejection Fraction from a Spanish Multicentre Study: SPANIK-HF Design and Baseline Characteristics"

_jcm, 2022, doi:10.3390/jcm11051170_

Round 1

Reviewer 1 Report

Jcm-1592633

Thank you for the opportunity for reviewing your article, “Prevalence, incidence, and outcomes of hyperkalaemia in patients with chronic heart failure and reduced ejection fraction from a Spanish multicentre study: SPANIK-HF. Design and baseline characteristics”

This is a design paper. Actually hyperkalemia is important for HF patients, even out patients.

#. However, I’m afraid that I don’t understand the purpose of this study.

Maybe you know, the clinical role of hyperkalemia in HF has already been resolved.

I think the merit of your study is “multicenter, prospective, observational” study in Spain.

Furthermore, the frequency of hyperkalemia is only 19.5%.

You described it is higher than previous study.

I’m not sure what is the special purpose of this study according to the paper.

You described as followings.

---prevalence, incidence, and prognosis of hyperkalaemia in patients with HF) was to estimate the prevalence and, on a medium-term basis (12 months), incidence of hyperkalaemia in HFrEF outpatients, and its relationship with non-optimal HF therapy and clinical outcomes (mortality and hospital admission).

You described as Nonetheless, there is some limited prospective evidence for the estimation of hyperkalaemia in HFrEF patients

What is “special”? Because we have already known the fact that hyperkalemia is prognostic factor.

#. In your discussion, you described previous articles about the relationships between cardioprotective agents and hyperkalemia.

However, I have not found any special, new information.

You should have described about the new information which would be obtained by your new study.

You should change your discussion in order us to understand the new information in comparison with the previous studies you described in the article.

For example, “In a recent observational study including patients initiating MRA therapy, the occurrence of hyperkalaemia led to MRA discontinuation in 47% and dose reduction in 10% of patients. Once MRA was discontinued, more than three-fourths of patients were not re-started on MRAs during the subsequent year”: What do you like to reveal the new information from your study?

You better describe the detail. Otherwise, we don’t obtain new information.

You should emphasize the merit of your study in each points.

#. EF, BNP, NT pro BNP are the only description about the disease.

We need more information. I think you better show the additional information, if possible.

#. We would be pleased to see the diagram in your paper.

Author Response

First of all, I would like to thank the Editor and reviewers for considering our manuscript "Prevalence, incidence, and outcomes of hyperkalaemia in patients with 
chronic heart failure and reduced ejection fraction from a Spanish 
Multicentre study: SPANIK-HF. Design and baseline characteristics" for its publication in Journal of Clinical Medicine.

We have thoroughly read all the comments and suggestions made by the reviewers. We aim to solve our manuscript's doubts and describe the amends we made following your recommendations with this letter.  

Reviewer 1."Thank you for the opportunity for reviewing your article, "Prevalence, incidence, and outcomes of hyperkalaemia in patients with chronic heart failure and reduced ejection fraction from a Spanish multicentre study: SPANIK-HF. Design and baseline characteristics". This is a design paper. Actually hyperkalemia is important for HF patients, even out patients.

Q1: However, I'm afraid that I don't understand the purpose of this study.

Maybe you know, the clinical role of hyperkalemia in HF has already been resolved. I think the merit of your study is "multicenter, prospective, observational" study in Spain. Furthermore, the frequency of hyperkalemia is only 19.5%. You described it is higher than previous study. I'm not sure what is the special purpose of this study according to the paper.

You described as followings. ---Prevalence, incidence, and prognosis of hyperkalaemia in patients with HF) was to estimate the Prevalence and, on a medium-term basis (12 months), incidence of hyperkalaemia in HFrEF outpatients, and its relationship with non-optimal HF therapy and clinical outcomes (mortality and hospital admission). You described as Nonetheless, there is some limited prospective evidence for the estimation of hyperkalaemia in HFrEF patients. What is "special"? Because we have already known the fact that hyperkalemia is prognostic factor."

Thank you for your comment, which reveals we failed to describe our objectives adequately.

Whereas the Prevalence, incidence and prognostic impact of hyperkalaemia in the heart failure population are well-known, our study focus on ambulatory patients treated only by heart failure specialists and whose heart failure treatment has been optimized to maximally tolerated doses according to current practice guidelines.

The role played by hyperkalaemia in this particular scenario, excluding patients undergoing drug titration, hospitalized patients, and patients with heart failure decompensation has not been defined.

For this reason, we decided to run a 1-year follow-up to determine:

1.- The incidence of hyperkalaemia during the first year after treatment optimization.

2.- The degree of compliance between heart failure specialists with clinical practice guidelines recommendations for estimating potassium levels during follow-up.

3.- The proportion of patients in whom medical treatment modifications were made due to the diagnosis of hyperkalaemia.

4.- The prognostic impact of hyperkalemia in patients in whom heart failure treatment optimization was accomplished.

Based on your concerns, we add the following quotes in the introduction section to better expose the rationale of our study:

"Nonetheless, there is some limited prospective evidence for the estimation of hyperkalaemia in HFrEF patients, especially in a recent cohort of well-treated patients in specialized units and once the treatment has been optimized.

So, the main objective of the SPANIK-HF study (Spanish multicenter study on Prevalence, incidence and prognosis of hyperkalaemia in patients with HF) was to estimate the Prevalence and incidence of hyperkalaemia during the first year of follow-up, its relationship with suboptimal treatment and clinical outcomes (mortality and hospital admission) in a recent cohort of HFrEF outpatients under optimized medical therapy.

This first article will address data from the reference population, Prevalence of hyperkalemia, and interference with optimization of medical treatment at enrollment."

 Q2 "In your discussion, you described previous articles about the relationships between cardioprotective agents and hyperkalemia. However, I have not found any special, new information. You should have described about the new information which would be obtained by your new study. You should change your discussion in order us to understand the new information in comparison with the previous studies you described in the article.

For example, "In a recent observational study including patients initiating MRA therapy, the occurrence of hyperkalaemia led to MRA discontinuation in 47% and dose reduction in 10% of patients. Once MRA was discontinued, more than three-fourths of patients were not re-started on MRAs during the subsequent year": What do you like to reveal the new information from your study?. You better describe the detail. Otherwise, we don't obtain new information. You should emphasize the merit of your study in each points."

Thank you very much for your valuable contribution. We have modified our discussion section according to your indications.

We have highlighted that our cohort exclusively includes patients in whom treatment had been considered optimized, which might explain the higher Prevalence of hyperkalaemia compared to other cohorts.

Likewise, we have also highlighted the interference between hyperkalaemia and optimal medical treatment at the time of inclusion in the study. When the 1-year follow-up is concluded, we'll analyze the dynamic changes of potassium levels and its prognostic consequences.

Q3 "EF, BNP, NT pro BNP are the only description about the disease.

We need more information. I think you better show the additional information, if possible."

We have added a brief quote to describe our population. We decided not to include the NYHA functional class since it is a not fixed variable throughout the follow-up.

As you can see in Table 1, we describe what we considered the most interesting variables. Anyway, in case you think other data we'll improve the description of our population, we'll be pleased to include them.

Reviewer 2 Report

The authors should include a section on concomitant use of patiomer and other potassium lowering drugs on management and outcomes of the Hfref patients

Author Response

Reviewer 2.- Authors should include a section on the concomitant use of patiomer and other potassium-lowering medications in the management and outcomes of patients with Hfref.

We thank reviewer 2 for his contribution. At the moment of patients' recruitment, these medications had not been approved, so no patients were under any of these treatments. Anyway, we have included a paragraph to describe the current availability of these drugs and its possible impact on heart failure patients.